# A Pectate Lyase Gene Plays a Critical Role in Xylem Vascular Development in *Arabidopsis*

**DOI:** 10.3390/ijms241310883

**Published:** 2023-06-29

**Authors:** Yun Bai, Dongdong Tian, Peng Chen, Dan Wu, Kebing Du, Bo Zheng, Xueping Shi

**Affiliations:** 1College of Horticulture, Jilin Agricultural University, Changchun 130118, China; yunb@jlau.edu.cn; 2College of Horticulture and Forestry Sciences, Huazhong Agricultural University, Wuhan 430070, China; 3Tobacco Research Institute, Chinese Academy of Agricultural Sciences, Qingdao 266101, China; 4College of Plant Science and Technology, Huazhong Agricultural University, Wuhan 430070, China; 5Poplar Research Center, Huazhong Agricultural University, Wuhan 430070, China

**Keywords:** *Arabidopsis*, pectin, pectate lyase, vascular development, xylem

## Abstract

As a major component of the plant primary cell wall, structure changes in pectin may affect the formation of the secondary cell wall and lead to serious consequences on plant growth and development. Pectin-modifying enzymes including pectate lyase-like proteins (PLLs) participate in the remodeling of pectin during organogenesis, especially during fruit ripening. In this study, we used *Arabidopsis* as a model system to identify critical *PLL* genes that are of particular importance for vascular development. Four *PLL* genes, named *AtPLL15*, *AtPLL16*, *AtPLL19,* and *AtPLL26,* were identified for xylem-specific expression. A knock-out T-DNA mutant of *AtPLL16* displayed an increased amount of pectin, soluble sugar, and acid-soluble lignin (ASL). Interestingly, the *atpll16* mutant exhibited an irregular xylem phenotype, accompanied by disordered xylem ray cells and an absence of interfascicular phloem fibers. The xylem fiber cell walls in the *atpll16* mutant were thicker than those of the wild type. On the contrary, *AtPLL16* overexpression resulted in expansion of the phloem and a dramatic change in the xylem-to-phloem ratios. Altogether, our data suggest that *AtPLL16* as a pectate lyase plays an important role during vascular development in *Arabidopsis*.

## 1. Introduction

Wood formation mainly involves three types of wall polymers: cellulose, hemicellulose, and pectin. All in all, the transcriptional regulation of secondary cell wall genes comprises large numbers of transcription factors, genes involved in the synthesis and assembly of the main chain and the side chain of different types of polysaccharides, as well as genes for the modification and degradation of polysaccharides [1]. Pectin is considered to be a major constituent of the dicot primary cell wall, but also an important component in secondary walls [2,3]. Pectin is abundant in soft tissues and cell types within the plant body, such as parenchyma cells, the middle lamella, and three-way junctions [4]. The presence of pectin not only introduces a “cusion” in between cellulose and lignin polymers that are considered to be more rigid, but the interconnection of pectin with hemicellulose and possibly lignin also increases the complexity of the wall polymer network.

There are three types of pectin polymers based on the sugar building blocks of the main polysaccharide chain: Homogalacturonan or HG, Rhamnogalacturonan I or RG-I, and Rhamnogalacturonan II or RG-II [5,6]. At least 67 different enzymes participate in the synthesis of pectin, including glycotransferases (GTs), methyltransferases (MTases), and acetyltransferases [4,7]. Enzymes involved in pectin degradation belong to several protein families, such as pectin esterases (PEs, EC 3.1.1.11), polygalacturonase (PG), and pectate lyase (PL). PEs include pectin acetyl esterase (PAE) and pectin methyl esterase (PME) that catalyze the removal of the acetyl and carboxyl group from homogalacoronans, respectively. PGs break down glycosidic bonds between galactose units. PLs depolymerize pectin by cleavage between α-(1,4) glycosidic bonds, resulting in unsaturated oligosaccharides [8,9].

Pectin synthesis is important at various developmental stages during plant growth. Pectin is needed for cell expansion, cell adhesion, tissue abscission, and pattern formation [10]. Although pectin is a minor component for the secondary cell wall, numerous studies have shown that disruption of pectin synthesis or modification also influences secondary wall formation [11,12]. During recent studies, several PMEs in woody plants have been shown to participate in the regulation of cell size, cell invasion, and tissue flexibility [13,14,15]. The formation of tension wood or compression wood requires different mechanisms for the deposition of pectin and pectin-related compounds [16,17]. Our previous study showed that receptor-like kinases in *A. thaliana* formed a co-expression network with *AtPLLs* [18], supporting *PLLs* as a member for regulating secondary cell wall formation.

Previous studies showed that *PLL* genes are members of the PL1 protein family (EC2.2.2.2), one of the many CAZY (www.cazy.org, accessed on 4 March 2021) family proteins related to carbohydrate synthesis [19,20,21,22,23]. *LAT56* and *LAT59* were the first *PLLs* identified from tomato [24], which had a high degree of amino acid sequence homology with PelC [25]. PelC was isolated from *Erwinia chrysanthemi*, a fungus responsible for soft rotting disease in plants [25,26]. The tomato *PLLs* were mostly expressed in mature pollen, whereas *PLLs* from other plant species were found to be expressed in the pistil [27,28], trachery elements [29,30], latex [31], phloem/xylem fibers [32], and ripening fruits [33]. Wehas has reported the identification of a total of 29 *PLLs* in *P. trichocarpa* [34]. *AtPLLs* were expressed in root, hypocotyl, and leaf veins [22,23], and the poplar *PLLs* were highly expressed in the vascular cambium and radiation expansion zone [34]. Overexpression of *PtPL1-18* resulted in lower pectin content and changes in the xylem structure, leading to easier destruction of wall polymers [34]. The recent reports have proven that the putative pectate lyase *PLL12* is necessary for long-distance phloem transport [35], and it is also required for growth of the vascular bundles in the *Arabidopsis* inflorescence stem. Although *PLL12* was expressed primarily in the phloem, it also affected cambium and xylem growth [36] and patterned the guard cell wall to coordinate the turgor pressure and wall mechanics for proper stomatal function in *Arabidopsis* [37].

In this work, we report the characterization of *PLL* genes in *Arabidopsis thaliana*. Based on xylem-specific expression, we chose four *PLLs* for a functional study using loss-of-function mutants as well as overexpression transgenic materials. Our results show that changes in the expression of certain *AtPLLs* influence the pectin content, but more interestingly phloem/xylem development. Our data suggest that *PLLs*, especially *AtPLL16* expressed in vascular tissues, play a critical role in pectin synthesis/remodeling and secondary cell wall formation during vascular development.

## 2. Results

### 2.1. Bioinformatics Analysis of the PLL Family Genes in Arabidopsis

A total of 26 *PLL* (pectate lyase-like) genes were identified in the genome of *Arabidopsis thaliana* [23]. The length of the AtPLLs ranged from 300 to 500 amino acids, with the molecular weight ranging from 32 kDa to 56 kDa (Appendix A). AtPLLs were divided into five groups based on their phylogenic tree analysis, and more than half of them were clustered into Group I (Figure 1A). All AtPLLs shared a conserved Pec_lyase_C (pelC) domain that is symbolic for pectate lyase activity (Figure 1A). Most AtPLLs contained a signal peptide on the N-terminus (Figure 1A), suggesting their secretion out of the cytoplasm for pectin degradation. Notably, our predictions for AtPLL15, AtPLL22, and AtPLL23 were not consistent with a previous publication from Sun et al. [23], possibly due to the different software parameters used. A significant amount of residue within the pelC domain was well conserved among AtPLL proteins (Appendix A). Four Asp residues from *Erwinia* PelC, corresponding to D206, D208, D231, and D235 in AtPLL16, were shown to be important for Ca^++^ binding (Figure 1B). Five residues, D222, H223, T274, P289, and R290, were involved in the binding of oligo-galacturonic acid substrates, and R287 was on the catalysis side. They were also retained in the AtPLLs (Figure 1B and Appendix A). The sequence and structure conservation of AtPLLs suggest that they might share similar functions in pectin remodeling/degradation.

### 2.2. In Silico Analysis for Selection of Xylem-Specific AtPLL Genes 

The global expression data of *AtPLLs* were downloaded from the *Arabidopsis* eFP browser (http://www.bar.utoronto.ca/efp/cgi-bin/, accessed on 10 July 2020), and a heat map was generated to compare the *AtPLL* expression between different tissues (Appendix A). The in silico analysis suggested three groups of *AtPLL* genes, with Group I genes highly expressed in mature pollen, Group II genes highly expressed in vascular tissues, such as hypocotyl, root, and stem/internode tissues, and Group III genes highly expressed in flowers (Appendix A). A subset of Group III genes (*AtPLL7*, *13,* and *17*) were also expressed in the vegetative and reproductive apex (Appendix A).

Absolute expression values from AraNet (http://aranet.mpimp-golm.mpg.de/aranet, accessed on 10 July 2020) revealed that nine *AtPLL* genes were the most highly expressed in the hypocotyl and first internode of the basal stem (Appendix A), including four prominent vascular-specific genes (*AtPLL15, AtPLL16, AtPLL19,* and *AtPLL26*) (Appendix A), suggesting that they might play a critical role for vascular development.

### 2.3. Knock-Out of Xylem-Specific AtPLLs Results in a Thinner Hypocotyl and a Drastic Loss of Trachery Elements, Xylem, and Interfascicular Fiber Cells

We ordered T-DNA mutants for *AtPLL-15*, *-16*, *-19,* and *-26* in order to study their potential function in pectin remodeling and SCW formation during vascular development. Except for *AtPLL16*, where the T-DNA insertion was located in the 5′ UTR region, all the other T-DNAs were inserted in the protein coding region (Figure 2A and Appendix A). The transcript level of the corresponding genes was investigated by RT-PCR, using three pairs of primers corresponding to the 5′, the 3′, and regions spanning the T-DNA insertion sites. We found that the level of 5′ transcripts of *AtPLL15* cDNA was not affected, but the 3′-part transcript was considerably reduced compared with the wild type, and also the transcript around the T-DNA insertion site (Appendix A, top panel). For *AtPLL16* and *AtPLL26*, both the 5′- and 3′-transcripts were downregulated (Figure 2A and Appendix A). For *AtPLL19*, the level of the 5′-transcript was reduced, but not the 3′-transcript (Appendix A, third panel).

To investigate the roles of these *AtPLLs* in vascular development, hypocotyl sections were made during the seedling stage to check for any defects in vascular development. No significant changes were observed in the vascular tissue morphology in all mutants (Appendix A). However, the *atpll16* mutant showed fewer numbers of rosette leaves at the bolting stage compared with the wild type (Figure 2B,C). Morphological observations of the *atpll16* mutant showed that the *atpll16* mutant displayed a reduction in plant height, hypocotyl diameter, and stem diameter and an increase in hypocotyl length compared with the wild type at the mature stage (eight weeks old) (Figure 2D–H, Appendix A). Since none of the *atpll15, atpll19*, or *atpll26* mutants showed any significant difference in the stem or hypocotyl width/length at the seedling stage or the mature stage, *AtPLL16* was selected for further study.

Further comparison results of the cross-section of the basal stem internode from 6-week-old seedings show that the *atpll16* mutant had fewer vascular bundles (Figure 2I,J) and fewer xylem fiber and interfascicular fiber cells compared with the wild type (Figure 2K,L). Furthermore, at least four layers of phloem cells in interfascicular bundles could be identified in wild-type plants, whereas almost none could be observed in the *atpll16* mutant (Figure 2M,N). Additionally, distorted ray cells could be observed in 3-week-old *atpll16* mutant hypocotyl sections (Figure 2O,P), suggesting that function-loss of *AtPLL16* leads to vascular developmental defects.

To verify that the above phenotypes were caused by the function-loss of *AtPLL16*, *AtPLL16pro*::*AtPLL16* lines were constructed, and the RT-PCR results show that the *AtPLL16* expression level of all complement lines was rescued to the wild-type level (Figure 3A), and line 2 and line 3 were selected for further study. Morphological phenotype analysis showed that the plant height and rosette leaf number had fully reverted to the wild-type status (Figure 3B,C). The cross-section of the basal stem internode from 6-week-old seedings showed that the vascular tissues of the *atpll16* mutant had largely reverted to the wild-type level (Figure 3), evidencing that the above phenotypes were indeed caused by *AtPLL16*.

### 2.4. The atpll16 Mutant Had More Pectin and a Thicker Secondary Cell Wall

We further determined the thickness of secondary walls from tracheary element (TE) cells, xylem fibers (XF), and interfascicular fibers (IF) using electro-scanning microscopy (Figure 4). In all three cell types, the thickness of the secondary cell wall from the *atpll16* mutant was almost twice as thick as that from the wild type (Figure 4A,B for TE and XF, C and D for IF cells, E). The wall polymer composition was determined using basal stem internodes from wild-type and *atpll16* mutant plants (Figure 4F). No significant difference could be observed in the content of cellulose, hemicellulose, or total lignin, but the level of soluble sugar was reduced in the *atpll16* mutant, concomitantly with a slight but significant increase in pectin (Figure 4F). Although the total lignin level was not affected, the level of acid-soluble lignin (ASL) was slightly increased in the *atpll16* mutant, and the opposite occurred for acid-insoluble lignin (AIL) (Figure 4F). Given the function of PLL proteins in pectin degradation, we would expect an increase in pectin in the knock-out mutant of *PLL* genes (Figure 4F), which was indeed the case. The change in pectin content was not as dramatic, possibly due to a redundant function of other *PLL* genes.

In summary, downregulation of *AtPLL16* reduced the plant height, hypocotyl width, and number of vascular bundles in basal stems. The xylem vessel cell arrangement was distorted, and the xylem fiber cell and tracheary element cell walls were thicker. An abundance of pectin was added in the *atpll16* mutant. 

### 2.5. AtPLL16 Overexpression Led to an Imbalance in Phloem Expansion at the Expense of Xylem Reduction

A construct carrying the *AtPLL16* promoter and the GUS reporter gene was transformed into Col.0 to investigate the tissue-specific expression of the *AtPLL16* gene during different growth stages (Figure 5). Consistent with the eFP data, we found that the *AtPLL16* transcript was mostly expressed in the vasculature system, including leaf veins, hypocotyls, and roots in young seedlings (Figure 5A). In mature plants, *AtPLL16* was expressed in leaf veins, flower petals, siliques, and cortex, phloem, and xylem fiber cells in stem and hypocotyl cross-sections (Figure 5B–F). This expression pattern is consistent with the functions of *AtPLL16*, supporting the above results. 

To further investigate the function of *AtPLL16*, a 35S promoter-driven gateway vector was used to generate *AtPLL16* overexpression transgenic material. RT-PCR results show that the expression level of *AtPLL16* in AtPLL16ox transgenic lines was upregulated significantly (Figure 6A). Two of the overexpression lines, ox-2 and ox-5, with a higher expression level of *AtPLL16* were selected for phenotype analysis. The morphological observation showed that AtPLL16ox transgenic plants had a slightly reduced plant height compared with the control (Figure 6B,C). An obvious phenotype for the AtPLL16ox plants was manifested by the dramatic change in xylem vs phloem regions in hypocotyl cross-sections (Figure 6D–F, Appendix A). In wild-type plants, the xylem–phloem width ratio was approximately 2.0, whereas in AtPLL16ox plants this ratio was reduced by ca. 50% percent (Figure 6F). On hypocotyl cross-sections, the xylem region in AtPLL16ox plants was dramatically reduced (Figure 6D,E, yellow arrow); on the contrary, the phloem region had significantly expanded (Figure 6D,E, red arrow). On stem cross-sections, the interfascicular fiber cells were more abundant in AtPLL16ox plants compared with the control (Figure 6G,H). *AtPLL16* overexpression also resulted in a mild increase in the stem and hypocotyl width, as well as the hypocotyl length, compared with the wild type (Figure 6I). These results are contrary to *atpll16* mutant phenotypes, indicating that *AtPLL16* plays a critical role in vascular development in *Arabidopsis*.

## 3. Discussion

In this study, we described the identification and characterization of pectate lyase genes in *Arabidopsis thaliana*, and how downregulation or upregulation of *AtPLL16* influences the pectin content in secondary cell walls and vascular tissue development. Considering the presence of 26 *PLL* genes in *A. thaliana* and the high degree of sequence similarity, there might be a functional redundancy within each group of *AtPLL* genes; for example, Group III, which are closely associated with vascular tissues. However, within each group, there are differences in the signal peptide, pI, molecular mass of the protein, and additional unique motifs. By tertiary structure modeling, we found that the amino acid residues required for Ca^++^ binding are all conserved, as are the five residues necessary for the binding of oligo-galacturonic acid, a mimic of the natural substrate. However, in AtPLL1 and AtPLL2, one of the conserved His residues was replaced by Arg. A similar Arg residue exists in DKS1, a pectate lyase from *Bacillus pumilus*, and this Arg was shown to be critical to catalysis [38]. Therefore, we hypothesize that this Arg in *AtPLLs* also plays an important role for the protein’s function.

Besides vascular-tissue-specific *PLLs*, another group of *AtPLLs* showed a high level of expression in flower and, in particular, pollen tissue. Therefore, *AtPLLs* regulate vegetative growth by influencing vascular system development, but also regulate reproductive growth by possibly affecting pollen dehiscence. In addition, *PLLs* have also been shown to be involved in fruit ripening, seed germination, and lateral root growth [23]. In all these events cooperative cell dehiscence is needed, and *PLLs* fit in to degrade pectin during this process.

The TargetP prediction found that the AtPLL16 protein was located in the endoplasmic reticulum and belongs to the signal peptide secretion pathway. *AtPLL16* was mostly expressed in the vasculature system, including leaf veins, hypocotyls, and roots in young seedlings. In mature plants, *AtPLL16* was expressed in leaf veins, flower petals, siliques, and the cortex and phloem in stem and hypocotyl cross-sections. This localized expression pattern is in agreement with the reported role of PLL12 in the phloem [36] and stomatal guard cells [37]. One of the major findings in this work, however, is the dramatic change in the xylem–phloem ratio upon overexpression of the *AtPLL16* gene in the Col.0 background (Figure 6). These data, together with the phenotype and pectin content changes in the *atpll16* mutant, suggest a proper amount of *AtPLL16* expression is essential for *Arabidopsis* growth and development. A small amount of *AtPLL16* is not sufficient for cambium activity and the differentiation of phloem fiber cells (Figure 2). Some research has raised the possibility that a cell-wall-derived signal produced by *PLL12* in the phloem regulates cambium and xylem development [36]. On the cellular level, a decrease in pectate lyase activity slightly increased the pectin and ASL content (Figure 4). The changes in the cell wall polymer and structure may result in a feedback signal that in return inhibits cell division and produces premature secondary cell wall thickening. It was also found that pectin degradation through PLL12 appeared to be subtle in quantitative terms, and we speculate that PLL12 may act as a regulator to locally remove homogalacturonan, thus potentially enabling further extracellular enzymes to access and modify the cell wall during sieve pore maturation [35]. However, overall plant growth was inhibited in the *atpll16* mutant, manifesting as a thinner stem and hypocotyl and a reduced plant height (Figure 2). A large amount of *AtPLL16* disturbs the balance of xylem and phloem cells that originate from the vascular cambium, resulting in a massive expansion of phloem and a concomitant inhibition of xylem growth (Figure 6). We know that cell differentiation from the vascular cambium into xylem or phloem cells needs to reach a balance [39,40], and such a dramatic change in the balance between phloem and xylem is detrimental to plant growth (Figure 6). Pectate lyase might function in wall integrity signaling, not only directly acting on cell wall mechanics but also via intracellular signaling pathways to influence cell pressurization, which ultimately drives plant vascular development.

In summary, although pectin is not a major component of plant secondary cell walls, which are modified during cell differentiation, pectin fragments might function as signals in cell wall integrity pathways to influence the secondary wall component. Our work provides evidence of the function of *AtPLL* genes for vascular development. *AtPLL* genes affect not just pectin, but also the differentiation of cambium, maintaining a balance from cambium to xylem and phloem. However, more work needs to be performed to illustrate how changes in pectate lyase activity affect cambium activity and cell differentiation. The unique regulation pattern for each *AtPLL* in the different groups also awaits further investigation.

## 4. Materials and Methods

### 4.1. Source Material and Growth Conditions

The *Arabidopsis* Columbia-0 ecotype (Col.0) was used in this study. T-DNA Salk lines for *AtPLL15* (Salk_140078C), *AtPLL16* (Salk_017336C), *AtPLL19* (Salk_054235C), and *AtPLL26* (Salk_147295C) were purchased from the ABRC (https://abrc.osu.edu/, accessed on 2 October 2020). *Arabidopsis* plants were grown in soil in a controlled growth chamber with the temperature set at 22 °C with a 16 h photoperiod, with a humidity level of 60% and the light intensity set at around 100 μmolm^−2^s^−1^.

### 4.2. Bioinformatics Analysis of Arabidopsis PLL Genes

Protein sequences of AtPLLs were downloaded from the TAIR website (www.arabidopsis.org, accessed on 2 October 2020). Multiple sequence alignment was performed by ClustalX 2.0 (https://www.ebi.ac.uk/Tools/msa/clustalw2/, accessed on 10 October 2020), and a non-rooted NJ tree was constructed with MEGA 5 software with bootstrap analysis performed by 1000 iterations [41]. The protein structure was constructed using IBS 1.0 software. SignalP (http://www.cbs.dtu.dk/services/SignalP/, accessed on 9 December 2020) [42] and TargetP (http://www.cbs.dtu.dk/services/TargetP/, accessed on 30 May 2023) [43] were used in the prediction of signal peptides and protein subcellular localization. SWISS-MODEL (http://swissmodel.expasy.org/interactive/, accessed on 9 December 2020) was used to predict the protein tertiary structure [44], and the respective PDB output files were analyzed using Swiss-Pdb-viewer [45]. 

Gene expression data were downloaded from the Arabidopsis eFP browser (http://bar.utoronto.ca/efp/cgi-bin/efpWeb.cgi, accessed on 10 July 2020) [46]. Expression data from different tissues were horizontally normalized before the generation of the heat-map using Multi Experiment Viewer 4.7.3 [47]. AraNet expression data (http://aranet.mpimp-golm.mpg.de/, accessed on 10 July 2020) [48] were used to confirm the expression of selected genes in vascular tissues. GeneCAT (Gene co-expression Analysis Toolbox, http://genecat.mpg.de/, accessed on 14 July 2020) was used to analyze gene co-expression [49]. 

### 4.3. Molecular Cloning Procedures

For the genotyping of various T-DNA mutants, genomic DNA was extracted using a QIAGEN Mag Attract 96 DNA Plant Core Kit (QIAGEN^®^ Agilent Technologies, Inc, Dusseldorf, Germany). The gateway vector pH2GW7 was used as a destination vector for overexpression, and pKGWFS7 was used as a destination vector for the promoter activity assay. A fragment carrying the 2112 bp promoter region of *AtPLL16* was used for the pmt-GUS assay. Constructs were transformed in Col.0 via agrobacterium C58 using the flower-dipping method with hygromycin (pH2GW7) or kanamycin (pKGWFS7) as a selection marker.

### 4.4. Sectioning of Arabidopsis Hypocotyl or Inflorescence Stem for Observation of Vascular Development

*Arabidopsis* seeds were sterilized and germinated in MS medium, and 10-day-old seedlings were used for hypocotyl sectioning. Agar-embedded samples were fixed with LR white and sectioned into 4 μm thick pieces using a Leica RM2265 microtome (Leica Biosystems Nussloch GmbH, Wetzlar, Germany). Slides were stained with toluidine blue before observation using an Olympus BX63 bright field microscope (Olympus Corporation, Tokyo, Japan). At later developmental stages, the hypocotyl or basal stem from 3- or 6-week-old plants was sectioned and stained with Technovit 7100 before LR white, in order to compare the vascular morphology between mutants and the wild type.

For secondary cell wall thickness quantification, the basal stem from 6-week-old plants was sampled and fixed with 2.5% (*w*/*w*) glutaraldehyde for 1 h at room temperature, washed with 100 mM phosphate buffer (pH 7.0), and fixed in 1% (*w*/*w*) osmium oxide for 2 h at room temperature. Samples were washed again with phosphate buffer, dehydrated with serial ethanol solution, and finally washed with 100% acetone twice for 20 min each time. Samples were infiltrated with 70% Spurr Resin (with 30% acetone) for 2–6 h, followed by 100% Spurr resin at 70 °C for 24 h. Around 80 nm thick sections were made using a Leica EM UC6 ultramicrotome (Leica Biosystems, Wetzlar, Germany). Sections were stained with acetyl uranyl and lead citrate for 20 min, respectively [50], before observation under an H-7650 transmission electro-microscope (Hitachi, Ltd., Tokyo, Japan). Images were taken with a Gatan 830 CCD camera (Gatan, Inc, Pleasanton, CA, USA) and analyzed with ImageJ software (version 1.44).

### 4.5. Promoter-GUS Activity Assay

Three individual T1-generation *AtPLL16pmt::GUS* transgenic plants were sampled for a 2-week-old seedling, a mature rosette leaf, a stem (1st internode, main floral inflorescence), and a hypocotyl cross-section, flower, and silique. GUS staining was performed according to Bai et al., 2017 [34].

### 4.6. RNA Extraction and qRT PCR 

Rosette leaves from 3-week-old plants were used for RNA extraction using a TIANGEN RNAPrep Pure Plant Kit (TIANGEN Biotech Co., LTD, Beijing, China). About 500 ng of RNA was used for cDNA synthesis using a PrimeScript^TM^ RT Reagent Kit (Takara BioTech Co., Ltd., Beijing, China), with a DNase I treatment to remove DNA contamination. Semi-quantitative RT-PCR was performed, and At5g60390 (*AtEf1a*) was used as a reference gene. For each gene of interest, three pairs of primers were used, corresponding to the region around the T-DNA insertion site (if applicable), the 5′-part of the cDNA, and the 3′-part of the cDNA. The sequence of semi-quantitative RT-PCR primers is shown in Appendix A.

### 4.7. Quantification of Wall Polymer Contents by Step-Wise Extraction

At the mature stage of plant growth, basal stem internodes were collected both for the *atpll16* mutant and the wild type. A total of 0.5–1 g of fresh sample was dried at 105 °C for 20 min and further in a 60 °C oven for 3–4 days, ground into powder, and passed through a 40-mesh screen. A total of 0.1 g of ground powder was used for step polymer fractionation according to Wu et al. [51], with the exception of lignin, where a 0.5 g sample was used. In short, soluble sugar was removed by potassium phosphate buffer, lipid was removed by chloroform–methanol (1:1, *v*/*v*) extraction, and starch was extracted with DMSO–water (9:1, *v*/*v*). Pectin was extracted by 0.5% (*w*/*v*) ammonium oxalate. Hemicellulose was extracted by 4 M KOH, and cellulose was extracted by 67% H_2_SO_4_ (*v*/*v*).

The amount of pectin, hemicellulose, or cellulose was calculated based on the pentose, hexose, or uronic acid released during the step-wise extraction using a colorimetric assay, with D-xylose, D-glucose, and GalA (galacturonic acid) as the standard curve, respectively [52]. A UV-VIS spectro-photometer (V-1100D, Shanghai MAPADA Instruments Co., Ltd., Shanghai, China) was used for absorbance measurements. In particular, for pectin quantification, 1.0 mL of the supernatant from the ammonium oxalate extraction was mixed with 5.0 mL of a sodium borate/sulfate solution in boiling water for 5 min, cooled to room temperature, and readings were taken at 520 nm. A total of 100 μL of 1.5 mg/mL meta hydroxyl benzene was added, the mixture was incubated for 5 min at room temperature, and again readings at 520 nm were taken. The difference between the two readings reflects the amount of uronic acid/pectin content [53].

Total lignin was quantified by a two-step acid hydrolysis method [52]. The acid-insoluble lignin (AIL) was calculated gravimetrically after correction for ash, and the acid-soluble lignin (ASL) was measured using UV spectroscopy.

### 4.8. Statistical Analysis 

At least three biological replicates were taken for each data point, duplicates were performed for each semi-quantitative RT-PCR reaction, and triplicates were performed for cell wall polymer determination. Data from biological replicates were subjected to an ANOVA test. * and ** indicate significance by Student’s *t*-test at *p* ≤ 0.05 and *p* ≤ 0.01, respectively.

## Figures and Tables

**Figure 1 ijms-24-10883-f001:**
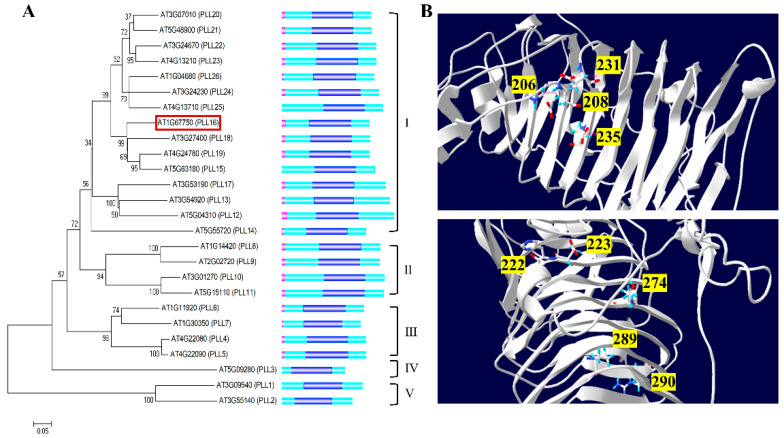
Phylogenetics and structure analysis of PLLs in Arabidopsis. (**A**) Phylogenetic tree and protein structure of AtPLLs. The phylogenetic tree was constructed by the neighbor-joining method with 1000 bootstrap replications using MEGA 5 software, and the alignment of the AtPLL protein sequences was performed using ClustalW. The numbers on the nodes indicate clade credibility values. AtPLL16 is highlighted with a red box. The protein structure of AtPLLs, constructed using IBS 1.0 software. The Pec_lyase_C domain is indicated with dark blue. The pink domain represents the predicted signal peptide. Clusters I–V on the right side indicate different clusters of PLL proteins according to Sun et al. 2010 [23]. (**B**) Tertiary structures of the AtPLL16 protein, predicted using the SWISS-MODEL online tool (http://swissmodel.expasy.org/interactive/, accessed on 10 March 2021). The numbers indicate four Ca^++^ binding sites (**above**) and five substrate binding sites (**below**).

**Figure 2 ijms-24-10883-f002:**
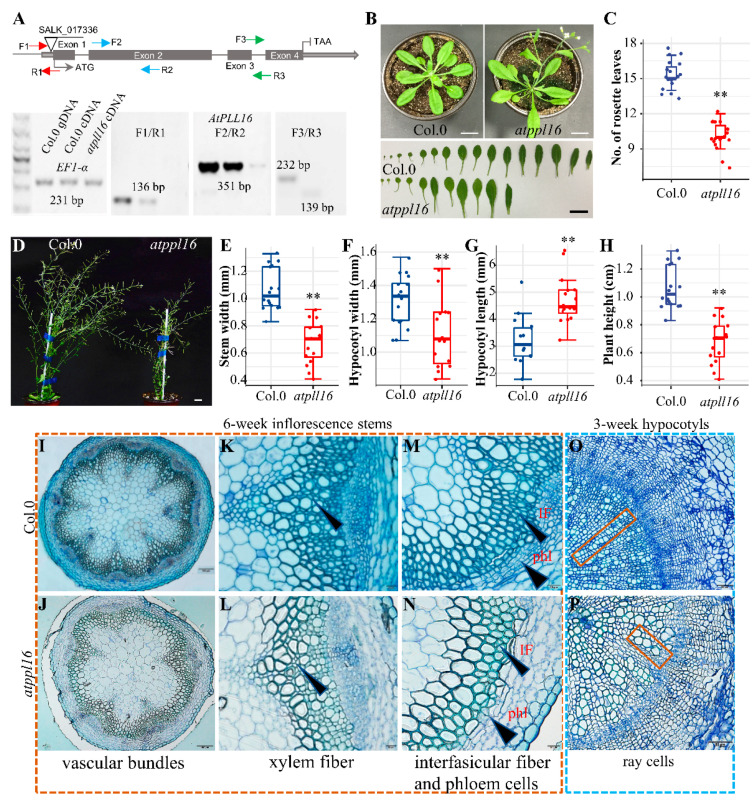
The *atpll16* mutant had vascular developmental defects. (**A**) The illustration for the T-DNA insertion of *AtPLL16* and RT-PCR results for *AtPLL16*. The triangle shows the insertion position and the lines on triangles represent the insertion direction. The arrows show three pairs of primers in the upper part. The results for the three pairs of primers are presented below, one spanning the T-DNA insertion site, the second corresponding to the middle part of the transcript, and the third corresponding to the 3′-part of the transcript. Col.0 gDNA and Col.0 cDNA were used as controls, *Ef1-α* was used as the reference gene in RT-PCR, and the DNA ladder is shown on the left in the lower part. (**B**) The 32-day-old Arabidopsis plants. (**C**) The numbers of rosette leaves from plants in (**B**). (**D**) The 8-week-old Arabidopsis plants. (**E**–**H**) Stem width (**E**), hypocotyl width (**F**), hypocotyl length (**G**), and plant height (**H**) of the *atpll16* mutant and wild type (Col.0) from (**D**), respectively. (**I**,**J**) Basal stem cross-section of the wild type and *atpll16* mutant; the *atpll16* mutant had fewer vascular bundles. Scale bar, 100 μm. (**K**,**L**) Zoom-in of the interfascicular bundle region of the wild type and the *atpll16* mutant. Arrows show xylem fibers. Scale bar, 20 μm. (**M**,**N**) Comparison of interfascicular phloem fiber cells of the wild type and the atpll16 mutant. Arrows indicate phloem cells and interfascicular fiber cells. Scale bar, 20 μm. (**O**,**P**) Hypocotyl cross-section of the wild type and the *atpll16* mutant. Orange boxes show ray cells. Scale bar, 50 μm. Sample size *n* = 16, ‘**’ indicates a significant difference at *p* < 0.01 by Student’s *t*-test. Bar = 2 cm in (**B**,**D**).

**Figure 3 ijms-24-10883-f003:**
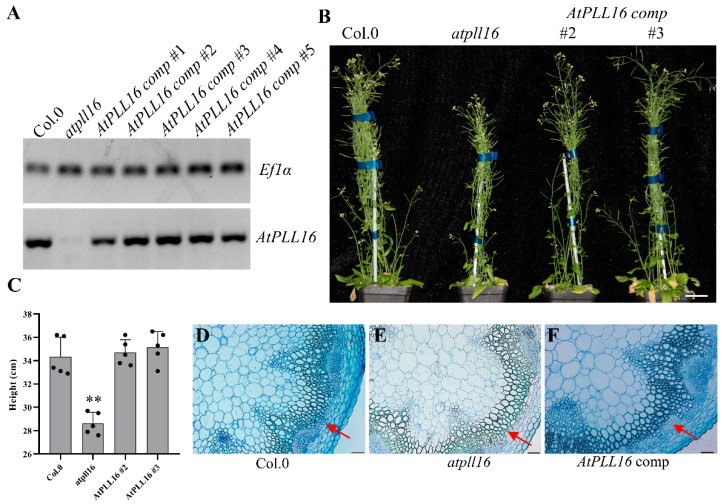
Complementation analysis of *AtPLL16* in the Arabidopsis *atpll16* mutants. (**A**) Characterization of the *AtPLL16* expression level in *AtPLL16* complementation transgenic lines. *Ef1-α* was used as the reference gene in RT-PCR. (**B**) Left to right, 8-week-old Col.0, *atpll16*, *AtPLL16* comp #2, and *AtPLL16* comp #3 plants. Bar = 5 cm. (**C**) Height of the plants in (**B**). Sample size *n* = 5, ‘**’ indicates significant differences at *p* < 0.01 by Student’s *t*-test. (**D**–**F**) Cross-sections of base inflorescence stems from 6-week-old Col.0 (**D**), *atpll16* (**E**), and *AtPLL16* comp #2 (**F**), representative. Bars = 50 µm. Red arrows represent phloem fibers.

**Figure 4 ijms-24-10883-f004:**
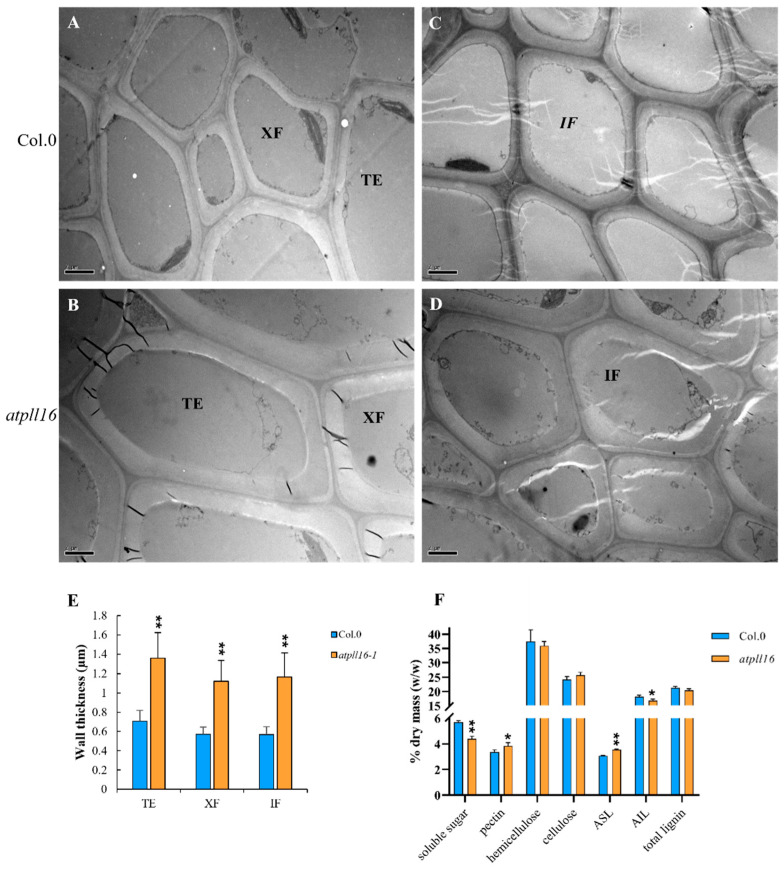
The *atpll16* mutant has more pectin and thicker secondary walls. (**A**–**E**) Basal stem from 6-week-old plants. (**A**,**B**) Xylem fiber (XF) cells and tracheary element (TE) cells in the wild type (**A**) and *atpll16* mutant (**B**). (**C**,**D**) Interfascicular fiber cells in the wild type (**C**) and *atpll16-1* mutant (**D**). Scale bar, 2 μm. (**E**) Wall thickness of TE, XF, and IF cells from the wild type and *atpll16* mutant. Data from at least 30 cells, ** indicates a significant difference at *p* ≤ 0.01 by Student’s *t*-test. (**F**) Quantification of wall polymers in basal stems at the mature stage. The abundance of each wall polymer is expressed as % drymass. ASL, acid-soluble lignin content; AIL, acid-insoluble lignin content. Data from three biological replicates and three technical replicates. ‘*’ or ‘**’ indicates a significant difference at *p* ≤ 0.05 or *p* ≤ 0.01 by Student’s *t*-test.

**Figure 5 ijms-24-10883-f005:**
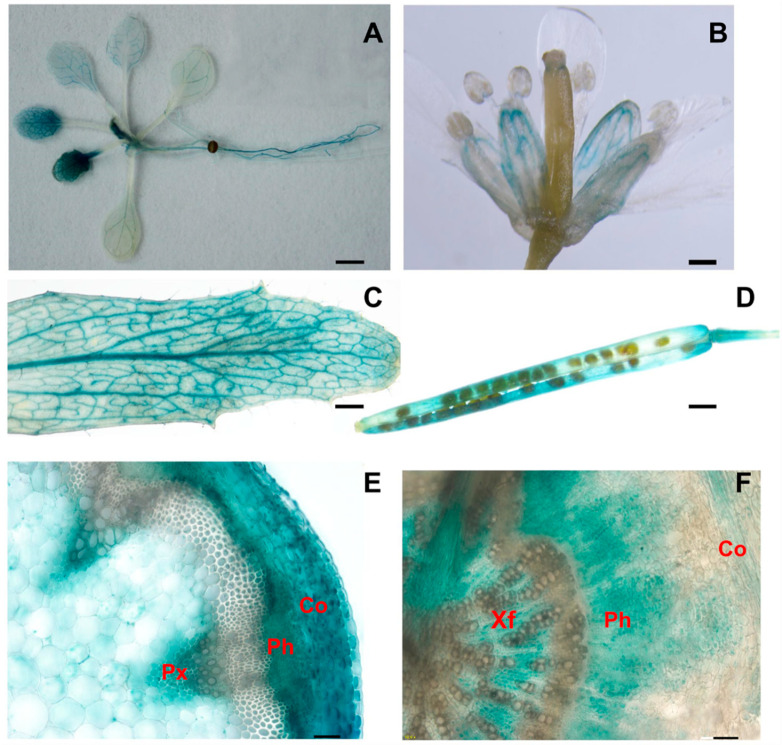
Promoter-GUS activity showing *AtPLL16* tissue-specific expression. (**A**) A 2-week-old seedling. (**B**) Floral organs. (**C**) Mature rosette leaf. (**D**) Silique. (**E**) Basal stem cross-section of 6-week-old plants. (**F**) Hypocotyl cross-section of 6-week-old plants. Co, cortex; Ph, phloem; Px, primary xylem; Xf, xylem fibers; (**A**–**D**) Bars = 1 mm; (**E**,**F**) Bars = 50 µm.

**Figure 6 ijms-24-10883-f006:**
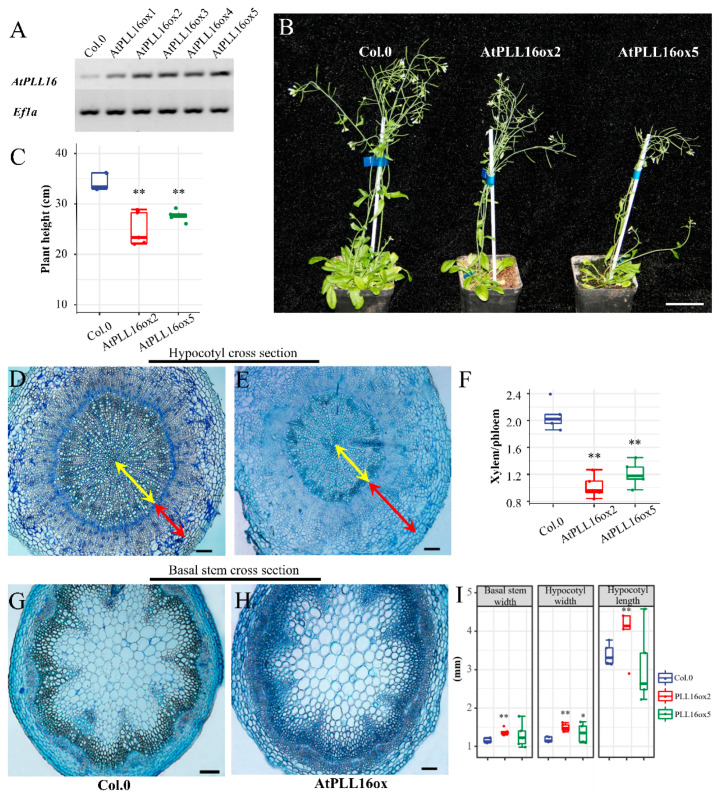
*AtPLL16* overexpression causes a dramatic change in the xylem–phloem ratio. (**A**) *AtPLL16* expression in transgenic plant leaves by RT-PCR, *Ef1-a* as the reference gene. (**B**,**C**) AtPLL16ox has a lower plant height at the mature stage (8 weeks old). Sample size *n* = 5, bar = 5 cm. (**D**,**E**) Hypocotyl cross-section of wild-type and AtPLL16ox plants (6 weeks old). Yellow arrows indicate xylem and red arrows indicate phloem. Scale bar, 100 μm. (**F**) Xylem–phloem width ratio of wild-type and AtPLL16ox plants. Sample size *n* = 5. (**G**,**H**) Basal stem cross-section of wild-type and AtPLL16ox plants. Scale bar, 100 μm. (**I**) Basal stem width, hypocotyl width, and hypocotyl length of wild-type and AtPLL16ox plants. Sample size *n* = 5. ‘*’ and ‘**’ indicate a significant difference at *p* ≤ 0.05 and *p* ≤ 0.01 by Student’s *t*-test.

## Data Availability

All data are presented in this article.

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
