# Peer review of "A Pectate Lyase Gene Plays a Critical Role in Xylem Vascular Development in Arabidopsis"

_ijms, 2023, doi:10.3390/ijms241310883_

Round 1
Reviewer 1 Report (New Reviewer)
I am positive about the manuscript, but it should be corrected for minor points concerning the English language.In the discussion, the authors could take into account the mechanical changes made to the cell wall and the potential involvement of wall integrity sensors when pectate lyases act on wall pectin. I am positive about the manuscript, but it should be corrected for minor points concerning the English language.
In the discussion, the authors could take into account the mechanical changes made to the cell wall and the potential involvement of wall integrity sensors when pectate lyases act on wall pectin.
Author Response
Please see the attachment.

Reviewer 2 Report (New Reviewer)
The paper presented by Yu Bai and colleagues is a very good example of combining both molecular biology techniques and plant anatomy. The authors showed evidence for the function of AtPLL genes for vascular development for a model species. It would be good to test the role of these genes in other species. The article deserves to be published but I have two comments:
Could the authors have used a double staining technique to better show the location of the xylem and phloem?
Please elaborate on the discussion of the role of pectins in plants in the vascular system and during changes in vessel formation.
Author Response
Please see the attachment.

This manuscript is a resubmission of an earlier submission. The following is a list of the peer review reports and author responses from that submission.
Round 1
Reviewer 1 Report
Despite recent reports on the biological functions of pectin modification, our understanding of its role in remodeling the cell wall is still limited. As a result, this paper's aim to identify and elucidate the function of PLL, which acts on vasculature formation, is of great importance. However, despite its intentions, the study's actual results were not entirely supportive. While attempting to elucidate the function of PLL16, the protein's localization did not be tested, and the mutant phenotype was not validated through complementation. Additionally, the data demonstrating expression patterns were of insufficient quality.
- Providing substantial characterizations, including confirming spatial and temporal expression patterns and protein localization, is necessary for the candidate PLLs. These fundamental pieces of information are crucial for interpreting the lack of phenotype in single knockouts, whether due to redundancy, differential spatial and temporal expression, or protein localization.
- I believe that additional experiments are necessary to understand the meaning of the phenotypes. To comprehend the biological function of PLLs, it is necessary to have experimental results demonstrating what changes occur in pectin and what role those changes play in actual material transport. Obtaining such results will enable a better understanding of the biological function of PLL16.
- To enhance the clarity of the figures, I suggest making a few modifications. Firstly, it would be helpful to indicate the area that has been zoomed in, as well as to clearly identify whether the figure shows WT or mutant data. Secondly, displaying actual data values rather than averaging them in a graph would be more beneficial, as this would enable readers to easily verify significance.
- The figure legends need to provide more accurate descriptions. It is important to specify how many samples were analyzed and provide clear explanations for any arrows or color labeling used in the figure. Many of these details are currently missing. The legend for Figure 6, in particular, contains several errors and requires further verification.
- There is a need to strengthen the references to recent studies on pectin and PLL-related research. (Examples: Chen et al (2021) Plant Cell 33(9):3134-3150; Bush et al (2022) Front Plant Sci 13:888201; Kalmbach et al (2023) Curr Biol 33(5):926-939.e9.)
Author Response
Dear reviewer,
We are really appreciate for your excellent and professional revision of this manuscript. We have checked the manuscript according to the comments. After carefully studying, we have made corresponding changes on the manuscript and uploaded to the attachment. Hope these will make it more acceptable for publication. Please see the attachment.
If any other information or modification are needed, please let me know, thank you so much.
Yours sincerely,
Yun Bai

Reviewer 2 Report
In their manuscript "A pectate lyase gene plays a critical role in xylem vascular development in Arabidopsis," Bai and colleagues report on the identification of PLL16 as a key regulator involved in xylem vascular development. The authors first characterized the expression of 19 PLL genes in Arabidopsis and selected the promising PLLs that were highly expressed in the hypocotyl and stem 1st internode. By observing the phenotype of the four pll mutants, the authors narrowed down PLL16 as an important regulatory pectate lyase as both pll16 mutant and overexpression of PLL16 induced the defects in the ratios of xylem to phloem, indicating the proper amount of AtPLL16 expression is essential for plant growth and development. All these results pointed towards that PLL16 is important for vascular development.
The authors explained the T-DNA mutants of the four PLL genes in lines 218-229, but it would be better to show the phenotypes of the other three mutants at the mature stages. This would enable readers to compare their phenotypes and understand the significance of PLL16's role in xylem vascular development. From Figure 3C, the phenotype of pll16 is not predominant, which seems to conflict with the main finding of the manuscript.
The figures need more information and changes to make the results more accessible to readers. In Figure 1A, it would be better to explain the representatives of the colors. In Figure 1B, the ID name on the left should be changed to the gene name instead. The legend for Figure 1C is missing, and the authors need to explain the structures of the upper and lower panels. It would be helpful to add important amino acids to the figure to clarify the domain structure.
In Figure 2A, the authors also observe high expression levels of other PLL genes, such as PLL1 and PLL18 from group II, in both hypocotyl and stem 1st internode. It would be better to show the absolute expression values of such genes, like in Figure 1B, to support the selection of PLL candidates.
In Figure 3, the illustration for T-DNA insertion is not commonly used, and it is difficult to determine the T-DNA insertion direction. It would be better to change it to a more common format. In Figure 3C, it is not clear which phenotype is caused by pll16, and pll19 appears to have a more obvious phenotype than pll16. The authors should explain this clearly in the manuscript.
In Figure 4, the authors only show the WT and pll16 mutants, but it would be better to include the complemented line in the same figure to indicate that the phenotype is indeed caused by the mutation of the PLL16 gene.
In Figure 5, the sample names in A-D should be labeled, and "Col-0" and "WT" should not be italicized in 1E and 1F. The significance of pectin and ASL is not clear from Figure 1F, so it would be better to add a broken axis for the Y axis.
In Figure 6B, the qRT-PCR illustrates five individual overexpression lines, but the other figures only show OE lines #2 and #5. It would be better to show the phenotype of all five lines in the same figure, unless the qRT-PCR only contains #2 and #5 without the results of other irrelevant lines. The sample names for D, E, G, and H should also be labeled.
Additionally, there are many typos in the manuscript. For example, in line 53, "SCW" should be written in full. "On the contrary" in line 21 should not be "one the contrary." "3-weeks-old" in line 137 should be "3-week-old." Genes and mutants should be italicized in many places (e.g. Line327; Line306; Line196; Line244, etc.).
Overall, it's easy for readers to read and understand the manuscript, but moderate editing in the English language will be helpful to improve the accessibility for a broad range of readers.
Author Response

(The authors gave the same response as above.)

Round 2
Reviewer 1 Report
Authors provided modified version of manuscript and figures. However, I still think that performing a phenotypic analysis using complementation lines or second allele of knockout and verifying protein localization are fundamental and critical steps for ensuring the accuracy and reliability of experimental results.
Author Response
Dear reviewer,
Thanks very much for your kind work and consideration on our paper.
- Sorry for not replying your question accurately in round 1. We had constructed complementation lines and analyzed their phenotypes. The results showed that AtPLL16pro::AtPLL16 lines reverted to wild-type phenotype, evidencing the experimental results were indeed caused by AtPLL16. This data was displayed in supplementary Figure 3 (please see the attachment) and were descripted in line 275-279.
- Thanks for your suggestion, we also agree that protein localization is critical for ensuring the accuracy and reliability of our results, but this experiment maybe spend two weeks in ideal conditions, beyond the review reply date, and we have no suitable tobacco seedlings to perform this experiment now, it will take more time. This experiment has been included in our future study. In addition, we had constructed the AtPLL16pro::GUS lines, and GUS report showed that AtPLL16 is expressed in vasculature system in tissues level, and xylem fiber cells and phloem cell in cell level, the results were showed in supplementary Figure 2 (please see the attachment), this expression pattern corresponds to the phenotypes of atpll16, which also supports our experimental results.
In addition, we have made further revisions to the manuscript.
If any other information or modification are needed, please let me know, thank you so much.
Yours sincerely,
Yun Bai

Reviewer 2 Report
Thanks for the careful responses and changes in the revised manuscript.
Author Response
Dear reviewer,
Thanks very much for your kind work and consideration on our paper. On behalf of my co-authors, we would like to express our great appreciation to you.
Thank you and best regards.
Yours sincerely,
Yun Bai
Round 3
Reviewer 1 Report
- Figure 1-3: Figure arrangement is required. Sine authors are only focus on the ATPLL16 and T-DNA insertion information and RT-PCR results for other genes are not necessary in the main figure. Figure 1 through Figure 3C can be reduced to a single picture, which will secure space for presenting more important results.
- Figure 3 (A-B): It would be helpful if the primer information (location and expected sizes) is displayed in the picture, and since RT-PCR results vary greatly depending on the primer set, it would be good to show the results under uniform conditions.
- Figure 3 (C-D): Plant ages and number of samples are missing in the legend, which is also missing in others (Figure 4, 5, and 6).
- Figure 4: Complementation phenotype is critical information to support reliability of the gene function. Therefore, the result (Supple Fig 3) has been presented at the main figure comparing with WT and atpll16 ko mutant. In addition, promoter GUS analysis (Supp Fig 2) results also have to be put in the main figure showing properly age dependent expression pattern of the genes especially at the hypocotyl and stem. In addition, protein localization analysis has to be shown to confirm their working position
- Figure 5 (E): Instead of expressing the bars in all different colors, it seems that the readability of the graph can be improved if WT and pll16 are displayed in the same color scheme but with a difference in brightness. Along with this, it would be good to unify the label color of the graph in the entire figure.
- Figure 5 (F): Full description of ASL and AIL are missing in legend. The number of test results should be indicated in the legend. In addition, a control that can prove whether this measurement result is reliable should be added. A mutant whose quantitative changes in pectin or other cell wall components have been verified in other papers may be a good control. It is also necessary to perform this experiment in complementation or overexpression lines.
- 3.4. atpll16 mutant had thicker secondary cell wall but more pectin: Sub-title linked with "but" are confusing, because increase of pectin in knock-out mutant of PLL genes was expected. Also, the atpll16 mutant was erroneously indicated as having a reduced amount of pectin (line 297). In the pll16 mutant, the thickness of the cell wall increased dramatically by about 2 times, but the quantitative change of pectin was not large, and the change of other cell wall components was also the same. An explanation of how PLL16 could affect cell wall thickness needs to be addressed.
- Figure 6: It is necessary to elucidate how cell wall thickness and composition of wall components behave in the overexpression line compared to the ko phenotype. The notation for the overexpression line needs to be unified, and the Y-axis label of graph (I) is missing. An accurate description of the samples (plant ages and tissues) used in RT-PCR is missing in the legend.
- Discussion part have to be improved.
Author Response
Dear reviewer,
Thanks very much for your kind work and consideration on our paper. We have made further revisions to the manuscript. Please see the attachment.
If any other information or modification are needed, please let me know, thank you so much.
Yours sincerely,
Yun Bai
